# Does Estimated Glomerular Filtration Rate Predict In-Hospital Mortality in Acutely Unwell Hospitalized Oldest Old?

**DOI:** 10.3390/geriatrics7060135

**Published:** 2022-11-28

**Authors:** Zack Robert Wakerly, Roy L. Soiza, Tiberiu A. Pana, Phyo Kyaw Myint

**Affiliations:** 1Ageing Clinical & Experimental Research Team (ACER), Institute of Applied health Sciences, University of Aberdeen, Aberdeen AB25 2ZD, UK; 2Aberdeen Royal Infirmary, National Health Service Grampian, Aberdeen AB25 2ZN, UK

**Keywords:** eGFR, oldest old, CKD, frailty

## Abstract

Globally the population of older adults is the fastest growing age group. Estimated glomerular filtration rate (eGFR) is an estimation of true kidney function with lower eGFR associated with higher mortality. However, few studies explore eGFR’s prognostic value in the nonagenarian. We investigated the association between eGFR on admission and mortality among the nonagenarians hospitalised with acute illness. A retrospective analysis of a prospective cohort study included patients aged ≥ 90 admitted into three acute medical assessment units or acute geriatric wards in England and Scotland between November 2008 and January 2009. Association between eGFR and all-cause mortality was evaluated using the Cox proportional hazard models controlling for potential confounders including frailty. 392 patients with mean (SD) 93.0 ± 2.6 years (68.45% women) were included. The median (IQR) eGFR was 26.61 (18.41–40.41) mL/min/1.73^2^. 63 died in in hospital. Low eGFR was not associated with mortality (Hazard ratio (HR) 1.00 (95% CI 0.98–1.02) overall or in sub–group analysis by frailty (HR 0.96 (0.92–1.01)) or by eGFR of ≤30 (HR 1.01 (0.95–1.06). We found no evidence of prognostic value of eGFR in predicting in–hospital mortality in the acutely unwell hospitalised nonagenarians.

## 1. Introduction

The population of older adults is the fastest growing age group [1] and is rising across the world. Age is associated with a decline in estimated glomerular filtration rate (eGFR), a marker of renal function, and therefore is also associated with chronic kidney disease (CKD). CKD is defined as an eGFR of <60 mL/min/1.73^2^ persisting 3 months or longer. Prevalence of CKD in older persons is high, exceeding 38% in community dwelling persons aged ≥ 70 [2]. Age related decline in renal function has multiple causes such as reduction in renal mass, glomerular sclerosis, tubular fibrosis, reduction in glomerular number, age related changes in haemodynamics and systemic hypertension [3,4,5].

The measurement of GFR can be directly or indirectly measured with various techniques. Direct measurement is practically very difficult to employ due to the rigid timing and a prolonged testing phase. An indirect method that is employed is the measurement of serum creatinine levels (Scr) however this has reliability issues in older people due to sarcopenia and depleting muscle stores of creatine, thus overall calculated eGFR is favored to determine kidney function.

Data from the United States Renal Data System in 2015 showed that among the 10.6% prevalence of CKD in individuals aged ≥ 65 years old only 1.8% of these individuals had no concomitant diabetes or cardiovascular disease, demonstrating that reduced eGFR is often associated with comorbidities. These co-morbidities include atherosclerotic cardiovascular disease, heart failure, hypertension, diabetes and cognitive impairment [6] which in turn may be fatal. Hence, low eGFR is associated with an increased risk of all-cause mortality as well as cardiovascular mortality [7]. However, a U-shaped relationship between eGFR and mortality has been reported with an eGFR of higher than 89 mL/min/1.73^2^ showing an increased risk of all-cause mortality in nonagenarians, although the same was not true for the more general older population [8]. Overall, there is limited literature on this population considering eGFR as a prognostic indicator in nonagenarians.

In this study we investigated the association between eGFR and mortality in the acutely unwell hospitalised oldest old patients in the United Kingdom (UK).

## 2. Materials and Methods

### 2.1. Study Population

Patient data were collected from 3 UK sites: Norwich and Norfolk University Hospital (NNUH), Aberdeen Royal Infirmary (ARI) and Woodend Hospital, Aberdeen (WH). The NNUH has a general catchment population of ~750,000 with ARI and WH having a combined catchment population of ~500,000. All patients aged ≥ 90 who were admitted to the acute medical assessment units (NNUH and ARI) or acute geriatric wards (WH) were included within the study. Patients who were admitted under non-medical teams such as surgical and Accident and Emergency were excluded from the study. Those who were admitted to a medical specialty other than geriatric medicine were also excluded. The data were collected prospectively in a 3-month period from November 2008 and all admissions during the study period were followed until hospital discharge, whereby the 1 November 2008 was the first admission date and the 24 December 2009 was the final discharge date.

### 2.2. Data Collection

The detailed description of study methods are published elsewhere [9]. In brief, the data were recorded at two-time points admission and discharge with demographic information (age, gender, etc.), residence prior to admission, presenting complaints, chronic co-morbidities, drug and social history, baseline observations and investigations, pre-morbid functional status (modified Rankin score) [9] and mobility status. Presence of co-morbidities was noted from medical records including correspondence, GP referral information and clinical history.

Prior treatment in the community by a GP was also noted along with details of any previous admissions within 1 month of the recorded hospital admission. No formal ethical approval was required as data were collected for a multicenter audit project and respective NHS Institutional Approvals were obtained.

### 2.3. eGFR

The Cockcroft-Gault modification of diet in renal disease (MDRD) equation was used to calculate eGFR, eGFR = 175 × (Scr)^−1.154^ × (Age)^−0.203^ × (0.742 if female) × (1.212 if Black). This equation was chosen because it was found that MDRD was more consistent over a 5-year period [10] with the smallest mean bias and highest accuracy in subjects with diabetes. From this, participants were stratified into two groups using eGFR cut off of 30 mL/min/1.73^2^ (≤30 mL/min/1.73^2^ and >30 mL/min/1.73^2^). This stratum was used in order to attempt to balance the distribution of the groups as there was a very large number of participants in the ≤30 mL/min/1.73^2^ group. Furthermore, the current literature uses the groups <30 mL/min/1.73^2^, 30–44.99 mL/min/1.73^2^, 45–60 mL/min/1.73^2^ and >60 mL/min/1.73^2^ to indicate different renal function categories and thus the groups were in line with previously adopted cut-offs.

### 2.4. Modified Frailty Index-5 (MFI-5)

MFI-5 was used as it has an equivalent predictive ability as the well-established MFI-11 [11] in older people. The MFI-5 score was then stratified into MFI-5 ≤ 1 and MFI-5 ≥ 2 in order to attempt to balance the distribution and show the difference between frail and non-frail which was adapted from the literature groups of MFI-5 = 0, MFI-5 = 1 MFI-5 ≥ 2.

### 2.5. Data Analysis

Statistical analysis was performed using SPSS version 27.0 (SPSS, INC., Chicago, IL, USA). Means (SD), medians (IQR), hazard ratio (HR) and 95% CI were calculated when relevant. Variables were chosen based on whether they have an effect as markers or direct effect on mortality. Patients that had incomplete, missing, or incorrectly entered data for the specified criteria of MFI-5 or eGFR were excluded.

Differences between groups were assessed using one-way ANOVA for parametric continuous variables, Kruskal–Wallis for non-parametric continuous variables and Chi-squared tests for categorical variables.

Multivariable Cox proportional hazards model were constructed to determine the independent contribution of eGFR (predictor/exposure) to the outcome of in-hospital mortality. The following models were constructed. Model (A) age and sex adjusted; (B) adjusted for age, sex and blood parameters (Haemoglobin, WCC, CRP, albumin, urea, creatine and sodium) (C) Age, sex, blood parameters and baseline observations (Systolic blood pressure and pulse) (D) Age, sex, blood parameters, baseline observations and comorbidities (E) Age, sex, blood parameters, baseline observations, comorbidities, and polypharmacy (defined as ≥5 medicines being taken) (F) Age, sex, blood parameters, baseline observations, comorbidities, polypharmacy, and MFI-5. There was further stratification to assess risk of frailty and eGFR for MFI-5 and eGFR strata using all the variables from model F. *p*-value of <0.05 were considered to be statistically significant.

## 3. Results

### 3.1. Patient Characteristics

Table 1 shows the descriptive data obtained as a whole cohort and as stratified into MFI-5 groups. Of the original 419 that had data collected, 27 were excluded for having missing information and or incompatible information regarding either mobility, date of birth, date of arrival or age. The mean (SD) age was 93.0 (2.6) years with 31.55% being male and 68.45% being female. Most patients, 237 (60.45%), had an eGFR ≤ 30 mL/min/1.73^2^ and 155 (39.54%) had eGFR >30 mL/min/1.73^2^. Most patients were non-frail with MFI-5 ≤ 1 = 292 (74.49%) and MFI-5 ≥ 2 = 100 (25.52%). A total of 63 (16.03%) participants died as in-patients. From this univariate analysis shown in Table 1, those with higher CRP, higher albumin, those with history of ischaemic heart disease (IHD), hypertension, diabetes and atrial fibrillation were more likely to have higher frailty.

#### 3.1.1. Multivariate Cox Proportional Hazard Models

The results of the multivariate Cox proportional hazard modelling are shown in Table 2. Lower eGFR was not associated with in-patient mortality in nonagenarians (aged 90 years and over) in any models.

Being frail or having increased levels of sodium, albumin and higher pulse showed an increased risk of mortality. Having hypertension or diabetes decrease the risk of mortality in the non-stratified models. In model B and C high sodium showed significance with a HR of 1.07 (95% CI 1.04–1.11) and 1.07 (95% CI 1.03–1.10) and a *p*-value of <0.001 and 0.001, respectively. In model D high sodium (*p*-value < 0.001), high pulse rate (*p*-value 0.018), hypertension (*p*-value 0.014) and diabetes (*p*-value 0.044) showed significance with HR of 1.08 (95% CI 1.04–1.12), 1.01 (95% CI 1.00–1.03), 0.48 (95% CI 0.30–0.90) and 0.13 (95% CI 0.02-0.98), respectively. In model E significance was seen in high sodium (*p* value < 0.001), high pulse (*p*-value 0.019), hypertension (*p*-value 0.012) and diabetes (*p*-value 0.038) with hazard ratios of 1.08 (95% CI 1.04–1.12), 1.01 (95% CI 1.00–1.03), 0.47 (95% CI 0.27–0.88) and 0.12 (95% CI 0.02–0.92).

#### 3.1.2. Stratified Analyses by Frailty

eGFR was not associated with mortality in nonogenerians regardless of frialty status.

However, in the non-frail nonogenerains (MFI-5 ≤ 1, Table 3) increased pulse rate showed a higher risk of mortality while having hypertension showing a reduced risk of mortality. In model G significant results included hypertension (*p*-value 0.009) with a HR of 0.33 (95% CI 0.15–0.76) and pulse (*p*-value 0.043) with a HR of 1.01 (95% CI 1.00–1.03).

In the frail nonogenerians (MFI-5 ≥ 2, Table 4) having diabetes showed a reduced risk of mortality. Model H showed diabetes (*p*-value 0.035) with a HR of 0.04 (95% CI < 0.01–0.80).

#### 3.1.3. Fully Adjusted Non-Stratified Cox Proportional Hazard Model (Model F)

Model F showed significance in high sodium (*p*-value 0.002), high pulse (*p*-value 0.026), hypertension (*p*-value 0.002), diabetes (*p*-value 0.015), and high MFI-5 (*p*-value 0.009) with HR of 1.06 (95% CI 1.02–1.10), 1.01 (95% CI 1.00–1.03),0.36 (95% CI 0.19–0.69), 0.02 (95% CI 0.01–0.62), 2.61 (95% CI 1.27–5.36), respectively (Table 5). However, as previsouly seen eGFR had no association with mortality.

## 4. Discussion

In this study we found no association between eGFR and all-cause mortality in acutely unwell hospitalized people aged ≥ 90 years in the UK NHS setting. As can be seen in Table 2 and Table 5 there were no significant results in any models of the Cox proportional hazard analyses even when categorised by frailty status. This indicates that eGFR does not have practical use as a prognostic indicator in the acutely unwell hospitalized nonagenarians.

The majority of patients were female with similar percentages across the two frailty groups. Furthermore, overall, there was a much lower number of frail patients compared to non-frail patients. There were higher incidences of IHD and stroke/CVD in the non-frail groups. Within the frail group there were higher incidences of hypertension, diabetes, COPD and atrial fibrillation (see Table 1).

This study’s results are contrary to the findings of Montesanto and colleagues [8] on 505 community dwelling subjects aged ≥ 90. BIS1 was used to calculated eGFR from which a crude and adjusted Cox regression analysis (adjusted for age, gender, and frailty).A hazard ratio for death of 1.53 (*p* = 0.028) was found for individuals with an eGFR of ˂30 mL/min/1.73^2^. This is diametrically opposed to the results found where no affiliation was discovered even when stratified for frailty (*p*-value > 0.05). In the aforementioned study the BIS1 equation was used which estimates lower GFR values compared to MDRD [12] and is not interchangeable with MDRD. Furthermore, our present study adjusted for a greater number of potential confounders suggesting that residual confounding in Montesanto and colleagues’ study may account for the difference in conclusion. Van Pottelbergh and colleagues in 2013 [2] further showed this in a Cox proportional hazard model on 378 88-year-olds with an eGFR slope per year of < −5 between 85 and 88 with < 45 mL/min/1.73^2^ (n = 62). When adjusted for gender and eGFR at age 88 a HR of 1.77 (0.52–6.10) for overall mortality and 4.00 (1.04–15.43) for CV mortality was found. This indicates increased risk of mortality with decreased eGFR but, similarly to previously mentioned studies the number of covariates adjusted for were small and there are large confidence intervals. There is further evidence however to suggest acute kidney injury and increased creatine levels are good prognostic indicators of long-term mortality in the patients with ST elevated myocardial infarction with cardiogenic shock [13,14]. Demonstrating that kidney dysfunction can cause mortality among the vulnerable older people. It is worth noting however these studies were conducted on patients with a mean age of 68.7 and 69.8 years old and that creatine was combined with age and ejection fraction to create an ACEF score [14] which predicted mortality. A strength of our present study is the large number of covariates adjusted for which may explain the difference in findings compared to the literature.

Another potentially important explanation is that the patients in our study presented with acute illness. Their acute pathology may be extra-renal in nature making eGFR non-discriminatory in this circumstance, so some deaths may have occurred in people with serious pathology but no renal problems. Furthermore, eGFR was calculated from initial blood tests thus patients with easily reversable renal pathologies may have been included in the statistical analysis affecting the strength of association with all-cause mortality.

Hypernatremia however showed consistent association with all-cause mortality (seen in Table 5) with a HR of 1.06 (95% CI of 1.02–1.10) showing its potential as a prognostic indicator. This coincides with the current literature as Liber, Sonnenblick and Munter in 2016 [13] showed in 33 > 70-year-olds hypernatraemic sodium levels of >150 mEq/L, had 58% mortality rate within 30 days compared to the 32% (*p*-value < 0.05). However, a small sample size was used potentially exaggerating the association although there are other studies including Madsen et al. 2015 [14] that demonstrate this in larger populations.

Surprisingly increased levels of albumin showed an increase in mortality however this may have been as a consequence of dehydration worsening their CKD rather than a direct action of albumin on mortality. It may also have been a chance finding.

### 4.1. Clinical Relevance of the Findings

This study found no clinical relevance or evidence to propose eGFR as a prognostic indicator in those aged 90 years or over (seen in Table 2). Frailty does not further appear to have any effect in this relationship with no-significant association between eGFR and mortality (Table 3 and Table 4). Thus, in totality this study suggests that eGFR does not have the same prognostic ability for all-cause mortality in nonagenarians admitted to hospital as in younger populations when considered in isolation. However, eGFR is still useful in diagnosing CKD and therefore is clinically relevant.

Sodium levels have evidenced association with all-cause mortality with this study finding no different. We further showed those with hypernatremia had a significant increased risk of mortality (seen in Table 5). Sodium is commonly measured and performed well as a prognostic indicator.

### 4.2. Limitations

Race was not recorded so adjustment for it was not possible during the calculation of eGFR thus potentially leading to under reported or over reported eGFR. However, the population aged 90 and over in Aberdeen and Norwich is overwhelmingly white Caucasians. There are some limitations related to the MDRD equation and thus the insignificant results found maybe as a consequence of this rather than non-association. Estimated GFR is less accurate in patients with AKI presenting a higher likelihood of survival. The MDRD equation was derived and validated in a cohort with few nonagenarians thus this potentially effects the accuracy within this population. The MDRD equation, similarly to all creatine-based equations, suffers from physiological variances in creatine including the reduction in older frail patients and conditions associated with reduced secretion. Furthermore, the equation was developed in a population with CKD and thus underestimation has also been highlighted as a limitation. This is not exclusive to MDRD as all the most common equations have quite significant limitations and therefore presents a wider problem requiring a need for more accurate and validated equations or biomarkers specifically designed for nonagenarians [15,16,17,18]. The result found regarding the role of albumin and increased mortality may have been a chance finding and thus must be interpreted with caution. Due to the collection being upon admission and singular there is no way to differentiate between patients presenting with AKI, CKD, or a combination of the two. With no way to differentiate, reversable pathology may have been considered during interpretation potentially increasing the chance of survival. We used already existing dataset and thus no formal sample size calculation was made. This might increase the risk of a type 2 error.

### 4.3. Direction of Further Research

This study shows the need for further research in this area. The recording of race is required in future studies as it modifies eGFR calculations, substantially effecting estimation. Moreover, a larger and prospective cohort study of patients ≥ 90 would be required to increase power and allow for specific study design and data collection whilst focusing on nonagenarians. Usage of all common equations combined with inulin clearance, would allow for validation of equations comparative to the gold standard. Furthermore, inulin clearance would validate GFRs association with all-cause mortality and thus clarify if eGFR is a clinically appropriate method for prediction. Ensuring repeated eGFR calculation over time, similarly to that of plasma clearance, would prevent the inclusion of easily reversable renal pathology allowing for the preventing of potential skewing. Analysis of results using trends would be important combined with Cox regression analysis to give a broader understanding of the relationship between eGFR and all-cause mortality. Additionally, the recording of relevant alternative end points including change in functional state and facility placement would be required to gain a more complete understanding of the role of CKD in age. Moreover, change in eGFR rather than low eGFR may cause the increase in risk of all-cause mortality. This was shown by Vart et al. 2019 [19] whereby stroke patients with low eGFR had substantially higher risk of all-cause mortality with a change of eGFR > 5 mL/min/1.73^2^ during the patients stay. This combined with our studies results warrant further investigation around the significance in change of eGFR.

## 5. Conclusions

In conclusion we have shown that eGFR was not associated with increased mortality in acutely admitted nonagenarians and therefore may not be a good prognostic indicator limiting its clinical relevance for prediction of mortality. This is contrary to the existing literature on older people in other settings where eGFR was shown to be a good prognostic indicator. Hypernatremia and change in eGFR show potential as better prognostic indicators however further elucidation on the clinical application within this population is required. There are still significant limitations around all existing eGFR equations for this population that must be addressed in future research to fully evaluate the clinical relevance of GFR. Whilst eGFR must be considered when understanding overall health and diagnosis upon admission, it may not be the most useful tool upon admission to predict underlying chances of mortality or general trajectory for nonagenarians.

## Figures and Tables

**Table 1 geriatrics-07-00135-t001:** Baseline characteristics of 392 older patients aged 90 years or over by MFI-5 categories.

	All	MFI-5 ≤ 1	MFI-5 ≥ 2	*p* Value
N	392	292 (74.49%)	100 (25.51%)	-
Age, Mean (SD)	93 (2.61)	93 (2.59)	93 (4.00)	0.19 ^a^
Sex, N (%)	M = 124	M = 89	M = 34	0.76 *
	(31.55%)	(30.48%)	(34.00%)	
	F = 269	F = 203	F = 66	
	(68.45%)	(69.52%)	(65.00%)	
Blood parameters on admission				
Haemoglobin Mean (SD) g/dL	12.20 (2.80)	12.46 (5.45)	12.20 (3.3)	0.79 ^a^
WCC Median (IQR) K/uL	9.6 (7.4–13.1)	9.4 (7.2–13.07)	9.95 (7.9–13.05)	0.16 ^k^
CRP Median (IQR) mg/L	78 (21.25–83.75)	78 (22–78)	78 (20.25–109)	0.033 ^k^
Albumin Mean (SD) g/L	37.00 (8.00)	36.03 (5.94)	34.00 (9.00)	0.02 ^a^
Urea Median (IQR) mg/dL	9.8 (7.2–13.9)	9.65 (7.03–13.75)	10.15 (7.5–15.1)	0.60 ^k^
Creatine Median (IQR) µmol/L	106.5 (81–136)	105.5 (80.25–134.75)	112 (84–139)	0.49 ^k^
Sodium Median (IQR) mEq/L	139 (135–141)	139 (135–141)	139 (135–142)	0.36 ^k^
Co-morbidities, N (%)				
IHD	144	110 (37.67%)	34 (34.00%)	0.029 *
Hypertension	184	106 (36.30%)	77 (77.00%)	<0.001 *
Diabetes	47	14 (4.79%)	33 (33.00%)	<0.001 *
Stroke/CVD	82	70 (23.97%)	12 (12.00%)	<0.001 *
Hyperlipidaemia	14	9 (3.08%)	5 (5.00%)	0.06 *
COPD	21	6 (2.05%)	15 (15.00%)	<0.001 *
Atrial fibrillation	80	55 (18.83%)	25 (25.00%)	<0.001 *
Baseline Measurements				
Systolic blood pressure Median (IQR) mmHg	136 (115–161)	137 (116–163.75)	131 (113–160)	0.57 ^k^
Pulse (Mean, SD) BPM	87.72 (22.11)	87.52 (22.72)	88.51 (21.33)	0.97 ^a^
Medications, N (%)				
Polypharmacy ≥ 5				
Yes	222 (56.49%)	159 (54.45%)	62 (62.00%)	0.10 *
No	171 (43.51%)	133 (45.55%)	38 (38.00%)	0.10 *
eGFR mL/min/1.73^2^, N (%)				
≤30	237 (60.45%)	175 (59.93%)	62 (62.00%)	0.34 ^k^
>30	155 (39.54%)	117 (40.07%)	38 (38.00%)	0.34 ^k^
Outcomes, N (%)				
Mortality	63 (16.03%)	46 (15.75%)	17 (17.00%)	0.035 *

^a^ = One-way ANOVA, * = chi-squared and ^k^ = Kruskal-Wallis test.

**Table 2 geriatrics-07-00135-t002:** Cox regression analysis results showing the association of eGFR as a predictor and all-cause mortality as outcome in oldest old adjusted for confounders and stratified by MFI-5 and eGFR using full model (Model F).

Model	Events (n)	HR for All-Cause Mortality for Low eGFR	95% CI	*p*-Value
A	63/392	1.01	0.99–1.03	0.38
B	63/392	1.00	0.98–1.02	0.80
C	63/392	1.00	0.98–1.02	0.96
D	63/392	1.00	0.98–1.01	0.64
E	63/392	1.00	0.98–1.02	0.70
F	63/392	1.00	0.98–1.02	0.79
Model’s stratified using MFI-5				
G (MFI-5 = ≤ 1)	63/392	1.01	0.98–1.03	0.67
H (MFI-5 = ≥ 2)	63/392	0.96	0.92–1.01	0.11
Model stratified using eGFR strata				
I (eGFR strata = 1 (≤30 mL/min/1.73^2^))	63/392	1.01	0.95–1.06	0.83
J (eGFR strata = 2 (>30 mL/min/1.73^2^))	63/392	1.02	0.98–1.05	0.35

Model (A) Adjusted for age, sex, eGFR. Model (B) model A and blood parameters. Model (C) model B and baseline observations. Model (D) model C and comorbidities. Model (E) model D and polypharmacy. Model (F) model E and MFI-5. Model (G and H) model E stratified into MFI-5 1 (≤1) and 2 (≥2). Model (I and J) model F stratified into eGFR 1 (≤30) and 2 (>30). Whereby eGFR was analysed as a continuous variable until model I. In model I and J eGFR was analysed in category. Model I use the eGFR reference value of >30 mL/min/1.73^2^ and model J used the reference value of ≤30 mL/min/1.73^2^. eGFR was utilized as a continuous variable up to model H and in model I and J the categories of ≤30 mL/min/1.73^2^ and >30 mL/min/1.73^2^ were used whereby for model I eGFR of >30 mL/min/1.73^2^ was utilized as the reference value and for model J ≤ 30 mL/min/1.73^2^ was used as the reference value.

**Table 3 geriatrics-07-00135-t003:** Fully adjusted multivariate Cox proportional hazards model for non-frail patients (MFI-5 ≤ 1).

MFI-5 ≤ 1	HR	95% CI	*p*-Value
Sex	1.29	0.49–3.40	0.60
Age	1.00	0.88–1.14	0.99
eGFR	1.01	0.98–1.03	0.67
Hb	1.02	0.99–1.04	0.27
WCC	1.00	0.97–1.03	0.88
CRP	1.00	1.00–1.01	0.31
Albumin	1.03	0.97–1.10	0.39
Sodium	1.05	0.98–1.13	0.15
Pulse	1.01	1.00–1.03	0.043
IHD	1.45	0.74–2.84	0.28
Hypertension	0.33	0.15–0.76	0.009
Diabetes	0.00	0.00–0.00	0.97
Stroke/CVD	0.70	0.28–1.74	0.44
COPD	0.85	0.11–6.63	0.87
Atrial Fibrillation	1.10	0.49–2.46	0.83
Polypharmacy	1.48	0.75–2.91	0.25

**Table 4 geriatrics-07-00135-t004:** Fully adjusted multivariate Cox proportional hazards model for frail patients (MFI-5 ≥ 2).

MFI-5 ≥ 2	HR	95% CI	*p*-Value
Sex	0.43	0.05–3.7	0.45
Age	1.05	0.79–1.41	0.74
eGFR	0.96	0.92–1.01	0.11
Hb	1.16	0.83–1.62	0.38
WCC	1.04	0.91–1.2	0.57
CRP	1.00	0.99–1.01	0.87
Albumin	0.98	0.88–1.09	0.69
Sodium	1.07	1–1.14	0.054
Pulse	1.00	0.98–1.04	0.77
IHD	1.34	0.33–5.50	0.69
Hypertension	0.35	0.83–1.62	0.19
Diabetes	0.04	0.00–0.8	0.035
Stroke/CVD	0.22	0.04–1.32	0.10
COPD	0.000	0.000–0.000	0.98
Atrial Fibrillation	0.44	0.09–2.30	0.33
Polypharmacy	1.54	0.31–7.72	0.60

**Table 5 geriatrics-07-00135-t005:** Fully adjusted multivariate Cox proportional hazards model F.

	HR	95% CI	*p*-Value
Sex	0.96	0.42–2.20	0.93
Age	1.00	0.90–1.12	0.97
eGFR	1.00	0.98–1.02	0.79
Hb	1.02	0.99–1.04	0.18
WCC	1.00	0.97–1.03	0.94
CRP	1.00	1–1.01	0.19
Albumin	1.01	0.96–1.06	0.66
Sodium	1.06	1.02–1.10	0.002
Pulse	1.01	1.00–1.03	0.026
IHD	1.46	0.82–2.60	0.20
Hypertension	0.36	0.19–0.69	0.002
Diabetes	0.08	0.01–0.62	0.02
Stroke/CVD	0.55	0.26–1.16	0.12
COPD	0.30	0.04–2.26	0.24
Atrial Fibrillation	0.92	0.48–1.78	0.81
Polypharmacy	1.40	0.77–2.55	0.27
MFI	2.61	1.27–5.36	0.01

The reference values for the above was the first to present unit increase hazard ratio, however it was then performed with values as a minus equivalent for, e.g., a value of 31.22 mL/min/1.73^2^ was converted to −31.22 mL/min/1.73^2^ to measure the hazard ratio of a unit decrease. Neither provided significant results with the minus conversion for eGFR presented.

## Data Availability

Data will be available (anonymized) upon reasonable request.

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
