# Peer review of "Does Estimated Glomerular Filtration Rate Predict In-Hospital Mortality in Acutely Unwell Hospitalized Oldest Old?"

_geriatrics, 2022, doi:10.3390/geriatrics7060135_

Round 1
Reviewer 1 Report (Previous Reviewer 1)
While the authors acknowledge several of the significant limitations of the study design in their correspondence with reviewers, this needs to be reflected in the paper itself under LIMITATIONS. At minimum they should 1) discuss that the data does not differentiate between AKI, AKI on CKD and CKD. 2) acknowledge the lack of a data regarding discharge creatine. 3) acknowledge the inaccuracy of eGFR in a patient with an AKI. 4) discuss in more detail the limits of the MDRD in this sub-population. In all cases they should state potential impact of 1-4 on the results and conclusions.
Section on clinical relevance of findings is not very compelling
Author Response
While the authors acknowledge several of the significant limitations of the study design in their correspondence with reviewers, this needs to be reflected in the paper itself under LIMITATIONS.
Response: We agree with your suggestion and have added the mentioned issues to the limitations section please see below.
- Discuss that the data does not differentiate between AKI, AKI on CKD and CKD.
Response: We agree with your suggestion and have added this into the limitations please see line 307 to 311.
- Acknowledge the lack of a data regarding discharge creatine
Response: We agree with your suggestion and acknowledgement of the lack of data regarding discharge creatine has now been covered in the limitation section as well as the result section. (Please refer to lines 260 to 265 and 309 to 311).
- Acknowledge the inaccuracy of eGFR in a patient with an AKI
Response: We agree with your suggestion and have now explicitly stated AKI rather than reversable pathology. (Please see line 295 to 297).
- Discuss in more detail the limits of the MDRD in this sub-population.
Response: We have now extensively covered the potential problems with the MDRD equation within the manuscript with the inclusion of aforementioned with which there is nothing else population specific to be said. (Please refer to lines 295 to 297, 299 to 301 and 307-311)
- Section on clinical relevance of findings is not very compelling
Response: We are sorry the reviewer feels the clinical relevance is not compelling. We know that nonagenarians are indeed survivors, and they have less physiological reserve due to the ageing process – therefore we believe findings which are unexpected for younger older patients are entirely plausible in this age group and indeed very relevant, important and novel.
Reviewer 2 Report (Previous Reviewer 3)
Authors clarified the requested points, so the paper is more clear.
Author Response
Authors clarified the requested points, so the paper is more clear.
Response: Thank you for your positive comment regarding our work.
Round 2
Reviewer 1 Report (Previous Reviewer 1)
The limitations section is much improved. However, the language is the newer portions could be more straightforward and concise. For example, in line 296, I would get rid of the "in general" so that it reads simply "estimated eGFR is less accurate in patients with AKI". I don't think it needs the caveat that "due to the nature of the pathology..." The sentence on line 309 "thus, reversable pathology" should also be reworded/edited for clarification
On line 299, I would put a comma after "the MDRD equation, similarly" and then again after the word equations.
Author Response
- The limitations section is much improved. However, the language is the newer portions could be more straightforward and concise. For example, in line 296, I would get rid of the "in general" so that it reads simply "estimated eGFR is less accurate in patients with AKI". I don't think it needs the caveat that "due to the nature of the pathology..." The sentence on line 309 "thus, reversable pathology" should also be reworded/edited for clarification.
Response: Thank you for your positive comments regarding our improved limitations section. We agree with all the changes suggested please see line 296, 309-311.
- On line 299, I would put a comma after "the MDRD equation, similarly" and then again after the word equations.
Response: We agree with your comments and see line 299 for the addition of commas.
This manuscript is a resubmission of an earlier submission. The following is a list of the peer review reports and author responses from that submission.
Round 1
Reviewer 1 Report
Given that this paper centered on hospitalized (acutely ill) patients, I think the prospect of acute kidney injury should have been factored in. Unfortunately this study did not include baseline creatine from the clinic setting and only admission and discharge values making it impossible to capture the prevalence of AKI. In the presence of AKI, eGFR if often a poor reflection of a patients renal function. From what I can tell the two other studies that found that eGFR was a significant predictor of mortality were based in the ambulatory care setting.
I agree with the authors that in the inpatient setting the reason for the hospitalization could also have been a very important factor to consider with respect to the primary endpoint but was not accounted for in this study.
It might have been nice to see other end-points besides all cause mortality that are important in this population, for example changes in functional status, the need for facility placement etc.
I think this study excluded ICU patients buts it is hard to tell. I also don't see mention or whether patients who are at baseline on dialysis or patients who require acute dialysis in the inpatient setting were included.
As the authors point out there is no perfect equation for measuring GFR, but it would be nice to understand why MDRD was chosen over CKD-EPI as well as Cystatin-C (which would have presumably would have been the best for patients with limited muscle mass)
While the population is stated to be overwhelmingly Caucasian based on the demographics of the locations used for the study, race should have been described. This is particularly true given recent controversy over the use of African-American race in calculations of eGFR and its impact on health disparities.
I could not find a reference to table 1 in the text of the article (? if I missed it)- might the authors want to comment on the relative homogeneity vs. heterogeneity in the baseline characteristics of patients included in this analysis.
Perhaps most importantly, the conclusion should speak more to the significance of the finding and its clinical implications. Even though there was no statistically significant results there is presumably still much to be learned. What key points should the reader take from the paper ?
Author Response
- Given that this paper centred on hospitalized (acutely ill) patients, I think the prospect of acute kidney injury should have been factored in. Unfortunately, this study did not include baseline creatine from the clinic setting and only admission and discharge values making it impossible to capture the prevalence of AKI. In the presence of AKI, eGFR if often a poor reflection of a patient’s renal function. From what I can tell the two other studies that found that eGFR was a significant predictor of mortality were based in the ambulatory care setting.
Response: We thank the reviewer for their insightful comment. We agree and thus we have mentioned this limitation of potentially including easily reversable renal pathology (please see line 257-258). We regret that we did not collect whether patient was presented with acute kidney injury, chronic renal disease or acute on chronic renal disease. We have included this as future direction of research in hospitalised extreme old age patient populations (see lines 304-307)
- I agree with the authors that in the inpatient setting the reason for the hospitalization could also have been a very important factor to consider with respect to the primary endpoint but was not accounted for in this study.
Response: Thank you for this useful and confirmatory statement with regard to our open and robust presentation of limitations of the study.
- It might have been nice to see other endpoints besides all-cause mortality that are important in this population, for example changes in functional status, the need for facility placement etc.
Response: We agree this certainly would have been a point of interest, unfortunately change in functional status wasn’t recorded and the discharge information was limited to place of residence and thus we were not able to include that information in the study. We highlight the need for future research regarding these other important outcomes in hospitalised older patients in this revised version (please see lines 319-321).
- I think this study excluded ICU patients buts it is hard to tell. I also don't see mention or whether patients who are at baseline on dialysis or patients who require acute dialysis in the inpatient setting were included.
Response: We apologise for the confusion however in a UK setting ICU admissions are few and far between in this age group. Even if this were to be the case they are usually admitted straight from accident and emergency which was not included. Furthermore, those who are admitted under geriatric care are not usually on dialysis and such details were not recorded in the study.
- As the authors point out there is no perfect equation for measuring GFR, but it would be nice to understand why MDRD was chosen over CKD-EPI as well as Cystatin-C (which would have presumably would have been the best for patients with limited muscle mass).
Response: Cystatin-C was not recorded routinely in clinical practice and thus we were not able to use it regarding GFR calculations. MDRD was chosen as both are widely used and considered to be accurate furthermore in the reference cited within the text (11); it was found that of the 5 tested, including CKD-EPI, MDRD showed significance for mortality in the earlier stages of CKD and thus the equation was used.
- While the population is stated to be overwhelmingly Caucasian based on the demographics of the locations used for the study, race should have been described. This is particularly true given recent controversy over the use of African-American race in calculations of eGFR and its impact on health disparities.
Response: We wholeheartedly agree that race should have been described however it was not recorded as part of the original data collection and thus couldn’t be included. Further data collection should include this however to attempt to address the health disparities that are faced by this population as GFR calculations suitability has been called into question.
- I could not find a reference to table 1 in the text of the article (? if I missed it)- might the authors want to comment on the relative homogeneity vs. heterogeneity in the baseline characteristics of patients included in this analysis.
Response: We apologise for this oversight, in the characteristics achieving statistical significance there was a larger percentage of incidences of IHD, stroke/CVD in the non-frail group. In the Frail group there was a higher incidence of Hypertension, diabetes, COPD, and atrial fibrillation. There overall was a lower number of frail compared to non-frail. This has been rectified within the text in this revised version (please see lines 234-238).
- Perhaps most importantly, the conclusion should speak more to the significance of the finding and its clinical implications. Even though there were no statistically significant results there is presumably still much to be learned. What key points should the reader take from the paper?
Response: We thank the reviewer for their useful comment. We believe that there are very relevant clinical implications and much to be taken from this. The key points taken should be that eGFR may not be a good predictor of mortality in this specific population, change in eGFR may be a better indicator, hypernatremia further showed very good potential, and that there should be further research to elucidate on this population We have amended the relevant texts to emphasise the nature of them being key points the readers should take from the paper.
Reviewer 2 Report
I have reviewed the manuscript entitled 'Does estimated glomerular filtration rate predict in-hospital mortality in acutely unwell hospitalized oldest old?'.
The manuscript appears to be interesting but several points should be changed before further consideration.
First, the phrase 'oldest old' is not a scientific definition and should be replaced with a more scientific one. ( octogenarian of etc)
The discussion appears to be very insufficient and should be improved since kidney functions has been documented to have prognostic role even in patients with myocardial infarction. Please state an explanation whether GFR has no role in endpoints in these patients. I recommend you consider citing 'Effect of acute kidney injury on long-term mortality in patients with ST-segment elevation myocardial infarction complicated by cardiogenic shock who underwent primary percutaneous coronary intervention in a high-volume tertiary center' and 'The predictive value of age, creatinine, ejection fraction score for in-hospital mortality in patients with cardiogenic shock'. The calculation of GFR may be the answer for the results of this study.
Author Response
I have reviewed the manuscript entitled 'Does estimated glomerular filtration rate predict in-hospital mortality in acutely unwell hospitalized oldest old?'.
The manuscript appears to be interesting, but several points should be changed before further consideration.
Response: Thank you for your positive comment regarding the interesting nature of our work.
- First, the phrase 'oldest old' is not a scientific definition and should be replaced with a more scientific one. (octogenarian of etc).
Response: We agree with your suggestion and have removed the term in favour of nonagenarian throughout the manuscript.
- The discussion appears to be very insufficient and should be improved since kidney functions has been documented to have prognostic role even in patients with myocardial infarction. Please state an explanation whether GFR has no role in endpoints in these patients. I recommend you consider citing 'Effect of acute kidney injury on long-term mortality in patients with ST-segment elevation myocardial infarction complicated by cardiogenic shock who underwent primary percutaneous coronary intervention in a high-volume tertiary center' and 'The predictive value of age, creatinine, ejection fraction score for in-hospital mortality in patients with cardiogenic shock'. The calculation of GFR may be the answer for the results of this study.
Response: We have added the aforementioned citation within the discussion as it raises a very interesting point. We believe that in the discussion we have outlined other potential possible explanations for the lack of association (please refer to lines 264-267,295-296 and 321-323) Furthermore, while it isn’t less useful in nonagenarians the results suggest that eGFR upon acute hospital admission isn’t a good predictor in this age group. We have further added the mentioned citations (please see line 255-261).
Reviewer 3 Report
The paper by Wakerly and coworkers may be of interest, unfortunately it shows relevant drawbacks.
First of all, it is not quite clear why Authors used MDRD and not other eGFR formulas more useful in older population as BIS1 or BIS2. Authors are kindly request to retest their data with different estimated GFR.
Moreover, because their enrolled pupulation with acute illness, eGFR need to be estimated also at discharge. It is well known that any eGFR formula is suitable only in steady-state conditions, and this is not the present case. Authors are requested to discuss these points.
It is quite surprising to found that high albumin level is associated with higher mortality: also this need to adequately discussed.
Last but not least, in this paper the enrolled population was very old, so it is quite clear that age is the main driver of mortality. In many other setting of geriatric population older than 65 years, eGFR was almost always related to mortality, although with lower powerful than frailty index.
I suggest to address all these points before revaluating the paper.
Author Response
- The paper by Wakerly and coworkers may be of interest, unfortunately it shows relevant drawbacks.
Response: Thank you for your comments. We have extensively discussed any shortcomings/ limitations of the work in the revised version and we sincerely hope that the reviewer will be supportive of the publication of the paper in the revised format.
- First of all, it is not quite clear why authors used the MDRD and not other eGFR formulas more useful in older population as BIS1 or BIS2. Authors are kindly requested to retest their data with different estimated GFR.
Response: MDRD was chosen as both are widely used and considered to be accurate furthermore in the reference cited within the text (11); it was found that of the 5 tested, including CKD-EPI, MDRD showed significance for mortality in the earlier stages of CKD and thus the equation was used. Thank you for your suggestion with regard to the additional analysis but we feel we have adequately justified the used of MDRD in this work which was supported by other expert reviewers.
- Moreover, because of the enrolled population with acute illness, eGFR needed to be estimated at discharge also. It is well known that any eGFR formula is suitable only in steady state conditions, and this is not the present case. Authors are requested to discuss these points.
Response: We agree and thus we have mentioned this limitation of potentially including easily reversable renal pathology (please see line 257-258). We regret that we were not able to confidently classify whether patient was presented with acute kidney injury, chronic renal disease or acute on chronic renal disease. We have included this as future direction of research in hospitalised extreme old age patient populations (see lines 304-307). Furthermore, it is unfortunate but this cannot be done as creatine levels were not recorded upon discharge and thus could not be calculated. Within the clinical setting in which this study was carried out, blood tests are not necessarily repeated either and therefore it would not be expected that everyone discharged will have creatine levels re-measured, meaning that such measurements are likely to be biased by the clinical need.
- It is quite surprising to find that high albumin levels are associated with higher mortality: also this need to adequately discussed.
Response: We agree with this comment it is surprising that this is the case and does warrant further comment. We have discussed the potential reasons for this (see line 274-276). We also think it could also be a chance finding- this has been added to the limitations (see line 302-303).
- Last but not least, in this paper the enrolled population was very old, so it is quite clear that age is the main driver of mortality. In many other settings of the geriatric population older than 65 years, eGFR was almost always related to mortality, although with lower power than frailty index.
Response: We agree that age is undeniably a factor when considering mortality, however the study was designed to specifically focus on the oldest old (nonagenarians). Whilst as seen in the more general elderly that eGFR is a good predictor of mortality the study was designed to find out if this association is still important to the very old which in turn is a different population and thus cannot be assumed to have the same association. Our findings therefore raise the possibility that not all prognostic factors for older people may be relevant to this patient population who are acutely unwell and thus provide new knowledge to clinicians.
- I suggest to address all these points before revaluating the paper.
Response: We thank you for the comments and the evaluation of out paper and hope the response provided will be satisfactory to the reviewer and that they would support the publication of the submitted work.
Round 2
Reviewer 1 Report
With respect to the new draft, line 258 says "within the vulnerable older people" and should be corrected for grammar.
Though the authors do a good job of pointing out the differences with their own research, I do not think the cited reference with regard to cardiogenic shock fits well in the paper. However, I understand this puts the study authors in a difficult position since article was explicitly recommended by another reviewer.
I am still concerned that the limits of the data collection as discussed in my prior review reduce the clinical significance of the article, however I'm not sure what the authors can really do any to address this retrospectively.
Author Response
- With respect to the new draft, line 254 says “within the vulnerable older people” and should be corrected for grammar.
Response: Thank you. This has been changed to “among vulnerable older people” see line 254.
- Though the authors do a good job of pointing out the differences within their own research, I do not think the cited reference with regards to cardiogenic shock fits well in the paper. However, I understand this puts the study authors in a difficult position since article was explicitly recommended by another reviewer.
Response: Thank you for the comment, we believe that these differences are very important to note when considering the broader context of the literature. We understand the concerns of the reviewer regarding the suitability of the cited reference however as noted due to the recommendation of another reviewer it has been included as part of this article.
- I am still concerned that the limits of the data collection as discussed in my prior review reduce the clinical significance of the article, however I’m not sure what the authors can really do anything to address this retrospectively.
Response: We agree there are limitations due to the retrospective nature of the analysis. We feel that we have highlighted these limitations adequately so that readers can make informed decisions in their clinical practice.
Reviewer 2 Report
Thank you for the required revisions.
Author Response
- Thank you for the required revisions
Response: Thank you for approving the revised version.